XA21-mediated resistance to Xanthomonas oryzae pv. oryzae is dose dependent

Zhang Nan 1 2
Dong Xiaoou 2 3 4 5
Jain Rashmi 2
Ruan Deling 2 5
de Araujo Junior Artur Teixeira 2
Li Yan 2 6
Lipzen Anna 7
Martin Joel 7
Barry Kerrie 7
Ronald Pamela C. pcronald@ucdavis.edu 2 4 5
1 Key Laboratory of Seed Innovation, Institute of Genetics and Developmental Biology, Chinese Academy of Sciences , Beijing , China
2 Department of Plant Pathology and the Genome Center, University of California , Davis , CA , USA
3 State Key Laboratory for Crop Genetics and Germplasm Enhancement and Utilization, Jiangsu Engineering Research Center for Plant Genome Editing, Nanjing Agricultural University , Nanjing , China
4 Innovative Genomics Institute, University of California , Berkeley , CA , USA
5 Feedstocks Division, The Joint Bioenergy Institute , Emeryville , CA , USA
6 Rice Research Institute and Key Lab for Major Crop Diseases, Sichuan Agricultural University , Chengdu , China
7 DOE Joint Genome Institute, Lawrence Berkeley National Laboratory , Berkeley , CA , USA
Domingues Douglas
Electronic publication date: 2024 May 6
Publication date: 2024
Volume: 12
Electronic Location ID: e17323
Received 2023 Sep 28; Accepted 2024 Apr 10
Copyright: ©2024 Zhang et al.
Copyright year: 2024
Copyright holder: Zhang et al.
License: This is an open access article distributed under the terms of the Creative Commons Attribution License, which permits unrestricted use, distribution, reproduction and adaptation in any medium and for any purpose provided that it is properly attributed. For attribution, the original author(s), title, publication source (PeerJ) and either DOI or URL of the article must be cited.
License URL: https://creativecommons.org/licenses/by/4.0/

Keywords: Rice, Plant defense, XA21, Receptor-like kinase, Genetic engineering, Xanthomonas oryzae pv. oryzae

Funding: Department of Energy Joint Genome Institute Office of Science of the U.S. Department of Energy DE-AC02-05CH11231 Chinese Scholarship Council Agricultural Science and Technology Innovation Program of Jiangsu Province CX(22)3153 The work (proposal: 10.46936/10.25585/60001021) conducted by the US Department of Energy Joint Genome Institute, a DOE Office of Science User Facility, is supported by the Office of Science of the US Department of Energy operated under Contract No. DE-AC02-05CH11231. Nan Zhang is also supported by a scholarship offered by the Chinese Scholarship Council. Xiaoou Dong is also supported by the Agricultural Science and Technology Innovation Program of Jiangsu Province [CX(22)3153]. The funders had no role in study design, data collection and analysis, decision to publish, or preparation of the manuscript.

==============================
The rice receptor kinase XA21 confers broad-spectrum resistance to Xanthomonas oryzae pv. oryzae (Xoo), the causal agent of rice bacterial blight disease. To investigate the relationship between the expression level of XA21 and resulting resistance, we generated independent HA-XA21 transgenic rice lines accumulating the XA21 immune receptor fused with an HA epitope tag. Whole-genome sequence analysis identified the T-DNA insertion sites in sixteen independent T0 events. Through quantification of the HA-XA21 protein and assessment of the resistance to Xoo strain PXO99 in six independent transgenic lines, we observed that XA21-mediated resistance is dose dependent. In contrast, based on the four agronomic traits quantified in these experiments, yield is unlikely to be affected by the expression level of HA-XA21. These findings extend our knowledge of XA21-mediated defense and contribute to the growing number of well-defined genomic landing pads in the rice genome that can be targeted for gene insertion without compromising yield.

Introduction

Genetic engineering enables the creation of desirable agronomic traits in crop plants by directly inserting genes of interest into the plant genome. The most common approaches for introduction of DNA into plant cells, Agrobacterium and particle bombardment-based transformation methods (Barton et al., 1983; Sanford, 1988), result in the insertion of genes at random sites in the target genome (Altpeter et al., 2016). Individual transformation events often exhibit fluctuations in their agronomic performance for at least two reasons. First, genes inserted at different genomic sites often exhibit difference in their expression level and pattern (Matzke & Matzke, 1998). Appropriate expression of the inserted genes of interest is often essential for the generation of the desirable traits in the engineered plants. Insufficient expression at the relevant tissues or developmental stages often results in the lack of the desirable traits, whereas excessive expression of transgenes can disrupt key biological processes and have a negative impact on growth or development (Rodriguez-Leal et al., 2019; Pino et al., 2007; Li et al., 2007; Hao et al., 2009). Secondly, DNA inserted at random genomic sites may affect the performance of the target plant by perturbing the expression of essential neighboring genes, as demonstrated by the study of the Golden Rice 2 event GR2R, where the insertion of the T-DNA disrupted an auxin transporter gene essential for the growth and development of the rice plant (Bollinedi et al., 2017). Due to the uncertainty associated with gene insertion at random locations in the genome during plant transformation, tremendous effort is often invested into examining numerous independent gene insertion events for their agronomic performance before they are considered for commercialization (Mumm, 2013). Therefore, it is desirable to identify genomic sites where transgenes conferring certain desirable traits can be safely inserted without incurring negative impacts on the recipient plant as candidate targets for targeted gene insertion approaches.

Plant immune receptors play crucial roles in the perception of infection-associated immunogenic signals and the initiation of defense responses (Zhou & Zhang, 2020). Cell–surface immune receptors and intracellular nucleotide-binding domain, leucine-rich-repeat containing receptors (NLRs) trigger robust immune outputs (Ngou, Ding & Jones, 2022). The rice XA21 gene, derived from the wild rice species Oryza longistaminata, encodes a plasma membrane-localized receptor kinase and confers broad-spectrum resistance to the rice bacterial blight pathogen Xanthomonas oryzae pv. oryzae (Xoo) when expressed in diverse rice cultivars (Song et al., 1995; Wang et al., 1996; Zhai et al., 2004; Tu et al., 1998; Zhai et al., 2000). Mechanistically, XA21 directly binds the sulfated peptide RaxX (required for activation of XA21-mediated immunity) secreted by Xoo and triggers defense signals leading to robust resistance (Pruitt et al., 2015; Luu et al., 2019).

To investigate the effect of genomic location and copy number of the XA21 transgene on the resistance conferred to the host, Zhai and colleagues analyzed independent XA21 transformation events generated in eight rice genetic backgrounds for copy number and location of the T-DNA inserts and assessed the resulting resistance to Xoo (Zhai et al., 2004). No obvious variation in the extent of Xoo resistance was observed among independent transformation events of the same genetic background, despite the difference in T-DNA copy number and the insertion sites. Nevertheless, since the expression of XA21 in these independent transformation events was not quantified, it remains to be determined whether a dosage effect exists between the amount of the XA21 protein and the resulting resistance to Xoo, as what has been reported for the NLR immune receptor SNC1 and the associated defense response in Arabidopsis (Xu et al., 2014). Zhai et al. (2004) reported differences in the level of Xoo resistance among XA21 transformants in different genetic backgrounds, suggesting cultivar-specific genetic interaction between XA21 and other rice genes.

Penalties on growth and development are often observed in plants with constitutively activated defense (Zhang et al., 2003; Li et al., 2007; Rodriguez-Leal et al., 2019; Sha et al., 2023). We examined the literature to determine whether such a correlation exist in the case of XA21. An XA21 transgenic event generated in the indica rice cultivar IR72 exhibited robust resistance to a wide range of Xoo isolates under field conditions without yield penalty (Tu et al., 1998; Tu et al., 2000). By contrast, severely decreased quality and yield was reported for an XA21 transgenic line in the japonica rice cultivar Taipei 309 (Hao et al., 2009). These reports suggest that distinct XA21 transgenic events under different genetic backgrounds may exhibit major differences in yield. However, because these studies were conducted by different groups and no quantification of the XA21 protein was reported, whether a negative correlation exist between the amount of XA21 protein and yield remains elusive. It would be desirable to identify genomic sites in rice where an XA21 transgene confers robust resistance with minimal yield penalty.

To determine whether the dosage of XA21 affects yield and resistance to Xoo resistance, it is essential to quantify the expression of XA21 and the level of resistance to Xoo in multiple independent transgenic events generated in the same cultivar. For this purpose, we generated thirty-nine independent XA21 transgenic events in the rice cultivar Kitaake. These plants accumulate XA21 protein at varying levels. We then measured Xoo resistance and key agronomic traits. We found a positive correlation between the amount of XA21 protein and the level of Xoo resistance. In contrast, we did not detect a correlation between the level of XA21 protein and the yield-related agronomic traits we quantified. These findings extend our knowledge on the regulation of XA21-mediated defense, provide useful germplasm for future investigation into the related defense signaling pathways, and identify promising genomic sites that can be utilized for engineered defense without affecting yield.

Materials and Methods

Rice lines used

Two rice lines used as the resistant controls are KitaakeX and XA21c4300. KitaakeX is a Kitaake rice line expressing an XA21 transgene driven by the maize Ubi-1 promoter (Park et al., 2008; Jain et al., 2019). XA21c4300 is a Kitaake transgenic line carrying the 9.8-kb XA21 genomic fragment with its native promoter as originally reported (Song et al., 1995; Peng et al., 2008). The robust resistance of KitaakeX and XA21c4300 to the Xoo strain PXO99 has been documented (Peng et al., 2008; Park et al., 2008).

Plasmid construction and rice transformation

Two sub-regions of the XA21 genomic sequence were individually PCR-amplified and joined by a second round of PCR to generate the full-length HA-XA21 fragment carrying the coding sequence of the HA epitope tag. The fragment was sequentially inserted into the pCambia2300 vector backbone using restriction enzymes to generate the pCambia2300-HA-XA21 construct. The HA-XA21 insert was verified by PCR-sequencing. The rice cultivar Kitaake was used to generate the HA-XA21 transgenic plants. Agrobacterium-based rice transformation was performed as described previously (Hiei et al., 1994) with modifications. Rice calli were induced by incubating mature rice seeds on the MSD medium (MS with 2 mg/L 2,4-D) for 2-4 weeks at 28 °C under a 16-hour light/8-hour dark regime. Calli were transformed by co-cultivation with the culture of Agrobacterium strain EHA105 carrying the pCambia2300-HA-XA21 construct for 30 min. Excess bacterium culture was removed by drying the calli on a sterile filter paper, after which the calli were moved to co-cultivation medium (MSD with 5% sorbitol and 200 µM acetosyringone) and incubated at 25 °C in the dark for 3 days. From this stage on, calli were kept separately to avoid the repeated counting of the transformants. After co-cultivation, calli were moved to the selection medium MSDH50 (MS with 2 mg/L 2,4-D, and 50 mg/L hygromycin) and incubated for 4-6 weeks. During selection, actively dividing sections of the calli were removed and place on fresh medium every 2 weeks. Surviving calli were transferred to regeneration medium (MS with 0.5 mg/L NAA, 3 mg/L 6-BA, and 50 mg/L hygromycin) and cultured under the same condition for an additional 4-6 weeks. Regenerating plants (T0 generation) were moved to MS medium for rooting. The rooted T0 seedlings were subsequently transferred to sandy soil in pots and were grown in tubs filled with fertilized water in a glasshouse as described before (Pruitt et al., 2015).

DNA extraction and genotyping

For genotyping, a segment of the leaf blade was harvested from individual rice plants. DNA extraction from the harvested leaf tissue was performed using a standard CTAB-chloroform-based method. PCR genotyping was performed using the Dreamtaq system (Invitrogen, Carlsbad, CA, USA). The sequences of the primers used can be found in Table S2.

Whole-genome sequencing analysis

For whole-genome sequencing, leaf segments were harvested from all tillers of a single T0 transgenic plant and combined. DNA was isolated from the harvested leaf samples and was used for library construction. Sequencing reaction was performed using the HiSeq 2500 sequencing system (Illumina, San Diego, CA, USA) at the Joint Genome Institute following the manufacturer’s instructions. To identify the locations of the T-DNA inserts in the T0 transgenic events, all sequencing reads carrying the 20-nt “probe” sequence GATCAGATTGTCGTTTCCCG, which is located near the right border of the T-DNA, were identified. A BLAST analysis was performed on each of these reads using the Nipponbare reference genome to locate the rice genomic sequence adjacent to the probe sequence. The locations of the genomic sequences are used to map the right boundary of the T-DNA insertions. For each T0 individual, the number of reads supporting each distinct insertion site was calculated. Accession numbers of the resequencing data in this study at NIH’s Sequence Read Archive (SRA) can be found in Table S3.

SDS-PAGE and western blot analysis

Total protein was extracted by incubating homogenized leaf tissue with the extraction buffer (PBS pH 7.2 with 0.15 M NaCl, 2 mM EDTA, 1% Triton X-100, 1 mM PMSF, 20 mM NaF, and 0.068% beta-mercaptoethanol) on ice for 30 min. Samples were incubated at 95 °C for 5 min in SDS loading buffer to denature, and were subject to SDS-PAGE according to Bio-Rad’s standard protocol. Western blot was performed using a mouse anti-HA primary antibody (Sigma H3663, 1:500 dilution: Sigma, Burlington, MA, USA) and a mouse IgGκ light chain binding protein (m-IgGκ BP) conjugated to horseradish peroxidase (Santa Cruz SC-516102, 1:1000 dilution) as the secondary antibody. Enhanced chemiluminescent assay was performed with the SuperSignal West Femto Maximum Sensitivity Substrate (Thermo, Waltham, MA, USA) using the Bio-Rad Gel Doc system.

Rice growth conditions and inoculation assay

Rice plants were germinated in tap water at 28 °C for 14 days before transplanted to sandy soil in 5.5-inch square pots (three seedlings per pot). After transplanting, rice plants were grown in tubs filled with fertilizer water in a glasshouse. Five to six weeks after transplanting, when the rice plants reach the booting stage, they were transferred to a growth chamber set to 28 °C/24 °C, 80%/85% humidity, and 14/10-hour lighting for the day/night cycle. Xoo strain PXO99 was cultured on PSA plates at 28 °C. Bacteria from the plates were resuspended in sterile water at a density of 108 cfu/mL and inoculated onto rice plants using the scissor clipping method as previously described (Kauffman et al., 1973).

Phenotypic analysis

Yield-related traits were quantified as previous reported with minor modifications (Zhou et al., 2023). Briefly, agronomical traits were assessed in ten-week-old homozygous T2/T3 plants under glasshouse conditions, including thousand-grain weight, grain length, grain width. Six randomly chosen plants were measured for each genotype. Statistical analysis was performed by using pairwise multiple comparison followed by Tukey’s test.

Results

Generation of XA21 transgenic events in the rice cultivar Kitaake

We based our design of the XA21 transgene on the previously reported 9.9-kb XA21 genomic fragment carrying its native promoter and terminator (Song et al., 1995). To enable the quantification of the XA21 protein, nucleotide sequence encoding three consecutive HA epitope tags was placed in-frame with the XA21 coding sequence near its N-terminus, between Arg80 and Val81, within the leucine-rich repeats region (Fig. 1A). This recombinant XA21 protein carrying the 3xHA epitope tag (named HA-XA21) is likely to maintain its normal function, because a Myc epitope tag inserted at the same position does not to interfere with the function of XA21, which has been documented in a previous report (Park et al., 2008). The HA-XA21 transgene was introduced into the rice cultivar Kitaake through Agrobacterium-based transformation (Hiei et al., 1994). Among the 39 independent T0 transgenic events we generated, the HA-XA21 fusion protein was detected in the leaf tissue in 28 independent T0 plants by western blot (Fig. 1B), indicating that the HA-XA21 transgene is expressed in most of the regenerated events. Because all T0 transgenic events were genotypically validated to be XA21 transgene-positive, we reason that those events where HA-XA21 was undetectable through western blot failed to accumulate the protein to a high level.

Figure 1 Expression of the HA-XA21 transgene in independent rice transformation events.

(A) Scheme of the T-DNA used for rice transformation. The T-DNA segment carries an HA-XA21 module (rectangle) and an NPTII selectable marker module (blue arrow). The amino acid sequence of the 3xHA epitope tag is displayed in detail. Full sequence of the binary vector can be found in the supplementary data. LB, left border; RB, right border. (B). Immunoblotting assay of the HA-XA21 protein in the T0 transformation events. Western blotting was performed using anti-HA antibody to detect the HA-XA21 protein in the 39 T0 transgenic rice plants. Total protein was extracted from the leaf blades of rice plants four weeks after the transfer of the rice plant to soil. Kitaake was used as the negative control. Red arrows indicate detectable HA-XA21 protein.

Independent transformation events harbor various copies of HA-XA21 at distinct genomic sites

To precisely locate the T-DNA insertion sites in the twenty-eight T0 events expressing HA-XA21, whole-genome sequencing was performed using genomic DNA from the individual T0 plants. For each T0 plant, sequencing reads that contain a 20-nucleotide sequence on the T-DNA near the right border were isolated. We expected most of these reads to also carry rice genomic sequence adjacent to T-DNA right border. By mapping these reads back to the rice genome, we identified T-DNA insertion sites in sixteen T0 events (Table S1). Each of the insertion sites are located at distinct regions randomly distributed in the rice genome. Ten events carry single-site T-DNA insertions. Five events had two insertion sites each. One event, 11A, carries insertions at five distinct sites. The distribution of the T-DNA copy number among the events we obtained is consistent with previously reported trends in Agrobacterium-transformed rice plants (Yang et al., 2005).

The amount of the HA-XA21 protein correlates with the level of Xoo resistance

To obtain homozygous lines of single-site HA-XA21 transgenic events, we planted T1 progeny derived from the six T0 events that carry the T-DNA insert at a single site in the genome based on the whole genome sequencing analysis described above (Table S1; 15A, 19A, 25A, 33A, 39A, and 47A). We then self-pollinated these T1 progeny and harvested T2 seeds from the T1 individuals. Because each of the T-DNA inserts contains an nptII selectable marker gene, which confers resistance to the selection agent G418, if the T2 progeny from a single T1 parent all exhibits resistance when germinated in the presence of 50 mg/L G418, this result indicates that the T1 parent carries a homozygous single-site insertion. Using this approach, we identified the T1 individuals 15A-1, 19A-2, 25A-1, 33A-5, 39A-4, and 47A-3 as being homozygous for the T-DNA insertion. We further validated the homozygous inserts in these lines by PCR using primers designed to test the presence and homozygosity of the T-DNA inserts based on the identified insertion sites (Fig. S1).

To quantify the HA-XA21 protein level, we performed a western blot assay and found that the amount of HA-XA21 protein accumulation differed in each of the transgenic lines (Figs. 2A, 2B). We next performed a leaf inoculation assay on these lines using Xoo strain PXO99 and observed various levels of resistance in distinct HA-XA21 transgenic lines (Fig. 2C). The rice plants used for the western blot and the Xoo inoculation assay were raised in parallel and were at the same age when the tissue harvest or the inoculation took place. To further control the experiment, the leaf segments harvested for total protein extraction correspond to the clipping site used in the Xoo inoculation assay. This experimental design helped ensure the biologically relevance of the tissue used for HA-XA21 protein quantification. We observed a positive correlation between the level of the HA-XA21 protein and Xoo resistance across the six transgenic lines (Figs. 2B, 2C), suggesting a possible dosage effect of XA21-mediated defense in the Kitaake rice cultivar. In particular, transgenic lines T0-25A-1 and T0-39A accumulated significant levels of HA-XA21 protein and conferred high levels of resistance approaching that conferred by transgenic Kitaake lines expressing XA21 under the maize Ubi-1 promoter (KitaakeX) or the native XA21 promoter (XA21c4300), both of which have been previously characterized (Peng et al., 2008; Park et al., 2008). Notably, KitaakeX constitutively expresses XA21 to a high level and displayed the strongest resistance to PXO99 in our inoculation assay, which is consistent with the hypothesized dosage effect of XA21. To validate that the HA-XA21 transgene is responsible for the Xoo resistance observed, we performed a co-segregation assay using the T1 progeny from T0-45A (Fig. S2). Co-segregation of the resistance and the HA-XA21 transgene was observed, which indicates that the HA-XA21 transgene accounts for the observed resistance to Xoo.

Figure 2 Accumulation of the HA-XA21 protein and Xoo resistance in transgenic events.

(A) Western blot assay of the HA-XA21 protein in transgenic offspring from nine transformation events using anti-HA antibody. The blue arrows indicate the six lines with homozygous single-site T-DNA insertions. Samples are total protein extracted from a segment of the flag leaf harvested five weeks after the transfer of the rice plants to soil. Coomassie Blue was performed in parallel to ensure equal loading. (B) Quantification of the intensity of the HA-XA21 band in (A) corresponding to the six homozygous lines. Numbers on the y-axis are arbitrary units. (C) Inoculation assay of the six homozygous lines using Xoo strain PXO99. Bars represent lesion length 14 days after clipping inoculation. Kitaake (white bar) was included as the susceptible control. Transgenic Kitaake lines expressing XA21 under the maize Ubi-1 promoter (KitaakeX) or the native XA21 promoter (XA21c4300) were included as the resistant controls.

All tested HA-XA21 lines display normal growth and development

To test whether the expression of HA-XA21 incurs any negative impact on the growth and development of the rice plants, we assessed the morphology of the rice plants and the panicles, and quantified several key yield-related traits including grain length, grain width, and 1000-grain weight of the six homozygous HA-XA21 transgenic lines (Fig. 3). We did not observe effects of the HA-XA21 transgene on plant stature, the size of the panicle, or the size and weight of the grains. These results indicate that XA21 expression in the transgenic lines we tested does not incur a yield penalty under our greenhouse conditions, making the corresponding insertion sites promising knock-in targets for engineered defense. For a more thorough analysis of agronomic traits of the above lines, field tests are needed.

Figure 3 Agronomic trait analysis of the homozygous HA-XA21 transformation events.

(A) Morphology of homozygous HA-XA21 transgenic plants ten weeks after transplanted to soil. Each pot contains three plants of the indicated line. All plants were grown in greenhouses under similar conditions. Kitaake and transgenic Kitaake lines expressing XA21 under the maize Ubi-1 promoter (KitaakeX) or the native XA21 promoter (XA21c4300) were included as controls. Bars represent 20 cm. (B). Upper: pictures of panicles from the plants in (A). Bars represent 5 cm. Lower: picture of grains from the plants in (A). Bars represent one cm. (C). 1000-grain weight, grain length, and grain width of plants in (A). Green bars correspond to plants harboring the XA21 transgene while the white bar represents the Kitaake control. Values are means ± SD. Different letters indicate significant differences ranked by pairwise multiple comparison followed by Tukey’s test (P < 0.05).

Discussion

Fine-tuning the expression of plant immune receptor genes holds the potential to maintaining a balanced growth-defense tradeoff. Studies on the NLR gene SNC1 in the model plant Arabidopsis have provided insights into the positive correlation between NLR expression and the robustness of the resulting defense responses. For example, epigenetic up-regulation of the expression of SNC1 is required for its function in plant defense (Li et al., 2011; Xia et al., 2013). Consistently, a dosage effect on the defense output was observed among independent SNC1 transgenic events with distinct expression levels (Xu et al., 2014). In the gain-of-function mutant snc1-1, a single amino acid substitution in the SNC1 protein increases its stability, resulting in an autoimmune phenotype, suggesting a positive correlation between the NLR protein and the intensity of the defense responses (Zhang et al., 2003; Cheng et al., 2011). Enhanced resistance due to increased SNC1 protein accumulation has also been reported in two additional mutants with defects in SNC1 turnover (Huang et al., 2014b; Huang et al., 2014a).

In contrast to the large number of studies on how NLR accumulation affects defense output and growth effects, there have been fewer reports on the relationship between the amount of cell-surface plant immune receptors and the resulting defense responses or corresponding yield. The expression of the Arabidopsis flagellin receptor gene FLS2 is positively regulated by the transcription factors EIN3 and EIL1 in an ethylene-dependent manner (Boutrot et al., 2010). Plants defective in ethylene signaling exhibited reduced FLS2 expression and increased susceptibility to the bacterial pathogen Pseudomonas syringae pv. tomato (Pst) DC3000 (Boutrot et al., 2010). Besides, transcription of FLS2 is repressed by the transcriptional regulators TOE1 and TOE2, which have overlapping functions (Zou et al., 2018). The toe1 toe2 double mutant exhibits increased response to flagellin treatment and enhanced resistance to Pst DC3000 (Zou et al., 2018). Two E3 ubiquitin ligases, PUB12 and PUB13, polyubiquitinate FLS2 and promote its degradation upon flagellin perception (Lu et al., 2011). The pub12 pub13 loss-of-function mutant exhibits reduced FLS2 turnover upon flagellin treatment as well as augmented defense responses (Lu et al., 2011). These studies suggest a possible dosage effect between FLS2 protein level and the robustness of the downstream defense. Since these studies were conducted in the model plant Arabidopsis, yield was not measured.

XA21 is among the first characterized cell-surface RLKs in plants (Ercoli et al., 2022), yet how the expression of XA21 quantitatively affects the function of the gene has not previously been well studied. Reducing the expression of an XA21 transgene by RNAi compromised XA21-medited immunity (Caddell et al., 2017), suggesting that XA21 expression level may influence the robustness of the resistance it confers. Constitutive activation of defense response can be detrimental to the growth and development of crop plants, which can also compromise their agronomic performances (Huot et al., 2014; He, Webster & He, 2022). To reduce the fitness cost incurred by “excessive immunity”, complex negative regulatory mechanisms are often adopted by plant hosts to restrict the activity of immune receptors (Couto & Zipfel, 2016). In the current study, we observed a positive correlation between the level of the XA21 protein and the extent of resistance to Xoo strain PXO99 in the Kitaake background, providing direct evidence of an XA21 dosage effect on resistance and no effect on yield (based on traits measured in the greenhouse). If a similar dosage effects is observed in future experiments with other cell surface immune receptors it would suggest a strategy to boost resistance without reducing yield.

It would be intriguing to explore whether the dosage dependency of XA21 is applicable to a broader range of rice cultivars including japonica rice and indica rice. Zhai et al. (2004) generated XA21 transgenic lines in three indica cultivars, and found that resistance level did not show any correlation with the copy number of the transgene, which implies that a dosage effect might be lacking in these cultivars. To verify this hypothesis, it will be necessary to generate additional indica transgenic plants and, obtain homozygous lines, and quantify the expression of XA21 as well as the defense levels in a comparative study.

Hao et al. reported in 2009 a sharp decrease in yield in a highly resistant XA21 transgenic line (Hao et al., 2009). We hypothesize that the decreased yield documented by Hao et al. is the result of a very high level of XA21 expression, based on the well-established defense-growth trade-off model. In contrast, in the current study, the native XA21 promoter was used to drive the transgene, which resulted in a moderate level of XA21 expression in all six lines examined. Insertion sites are also known to affect the expression of a transgene, which is often referred to as the position effect. This may account for the variation in the expression of the HA-XA21 transgene among distinct transformation events observed in our study. Event 25A exhibits the highest level of HA-XA21 expression with the most robust defense without an observed yield penalty. There is good evidence to suggest that the expression level of a transgene can be partly modulated by selecting an appropriate insertion site (van Leeuwen et al., 2001; Dean et al., 1988). The rapid development of genome editing tools has enabled efficient knock-in of large DNA fragments in plants (Lu et al., 2020; Joshi et al., 2020; Anzalone et al., 2022; Wang et al., 2022; Dong et al., 2020; Sun et al., 2023). Therefore, the T-DNA insertion sites reported in this study may serve as T-DNA insertion targets to introduce desirable traits into rice plants while maintaining their agronomic performance.

Conclusions

By analyzing six homozygous XA21 transgenic rice lines carrying single-site T-DNA insertion at distinct genomic sites, we discovered a positive correlation between the dosage of XA21 and the extent of the resulting resistance to Xoo. Examination of the agronomic traits of these lines under glasshouse conditions revealed that their grain yield is comparable to that of the background rice cultivar Kitaake. Overall, these findings extend our knowledge on the dosage effect of cell-surface immune receptors in plants for their better deployment in breeding, and contribute to the growing number of well-defined genomic landing pads in the rice genome for targeted gene insertion without compromising the performance of the resulting crop plants.

Supplemental Information

Supplemental Information 1 Genotypes determination of homozygous HA-XA21 lines

(A) PCR was performed with a pair of primers (LP and RP as in Table S2) that recognize genomic DNA flanking the T-DNA insertion sites on the T1 progeny of T0 events 15A, 19A, 25A, 33A, 39A, and 47A. These primers do not yield amplification product if the T-DNA insertion is homozygous as illustrated in the diagram on the top. (B) For each of the six indicated plants, two sets of primers are used to validate the genotyping results. Primer Set 1 recognizes the XA21 genomic sequence within the T-DNA and a primer that recognizes the genomic DNA near the inserted T-DNA right border (XA21Seq14F as in Table S2). Primer Set 1 was used to test the presence of the T-DNA at the expected insertion site in that line. Primer Set 2 is the same as the primers used in (A), which will not yield any amplification product only if the T-DNA insert is homozygous. T, transgenic sample; K, Kitaake control sample.

Supplemental Information 2 Co-segregation of the HA-XA21 transgene and the Xoo resistance

Inoculation assay of eight T1 progeny of the T0 event 45A using Xoo strain PXO99. Bars represent lesion length 14 days after clipping inoculation. White bars represent non-transgenic individuals. Filled green bars represent transgenic individuals. Kitaake was included as the susceptible control. A transgenic Kitaake line expressing XA21 under the maize Ubi-1 promoter (KitaakeX) was included as the resistant control.

Supplemental Information 3 Characteristics of the T-DNA insertions in 28 T0 plants expressing HA-XA21

Supplemental Information 4 Primers used in this study

Supplemental Information 5 Accession numbers of the resequencing data for 28 individual HA-XA21 T0 transgenic plants

Supplemental Information 6 Full sequence of the pCambia2300-HA-XA21 construct

Supplemental Information 7 Raw data for Figs. 2C, 3C, and Fig. S2

Supplemental Information 8 Uncropped gel images

We thank Mr. Phat Duong and Ms. Maria Hernandez for their technical support.

Additional Information and Declarations

Competing Interests

Author Contributions

DNA Deposition

Data Availability

Pamela C. Ronald serves as an Academic Editor at PeerJ.

Nan Zhang conceived and designed the experiments, performed the experiments, analyzed the data, prepared figures and/or tables, authored or reviewed drafts of the article, and approved the final draft.

Xiaoou Dong conceived and designed the experiments, performed the experiments, analyzed the data, prepared figures and/or tables, authored or reviewed drafts of the article, and approved the final draft.

Rashmi Jain analyzed the data, authored or reviewed drafts of the article, and approved the final draft.

Deling Ruan performed the experiments, authored or reviewed drafts of the article, and approved the final draft.

Artur Teixeira de Araujo Junior performed the experiments, authored or reviewed drafts of the article, and approved the final draft.

Yan Li performed the experiments, authored or reviewed drafts of the article, and approved the final draft.

Anna Lipzen performed the experiments, authored or reviewed drafts of the article, and approved the final draft.

Joel Martin performed the experiments, authored or reviewed drafts of the article, and approved the final draft.

Kerrie Barry performed the experiments, authored or reviewed drafts of the article, and approved the final draft.

Pamela C. Ronald conceived and designed the experiments, authored or reviewed drafts of the article, and approved the final draft.

The following information was supplied regarding the deposition of DNA sequences:

Accession numbers (28 in total) of the resequencing data in this study at NIH’s Sequence Read Archive (SRA) can be found in Table S3.

The following information was supplied regarding data availability:

The resequencing data for 28 individual HA-XA21 T0 transgenic plants is available at GenBank: SRP296494, SRP296493, SRP296492, SRP296490, SRP296489, SRP296488, SRP296487, SRP296485, SRP296484, SRP296483, SRP296482, SRP296481, SRP296480, SRP296479, SRP296478, SRP296477, SRP296476, SRP296475, SRP296474, SRP296473, SRP296471, SRP296470, SRP296469, SRP296468, SRP296467, SRP296466, SRP296465, SRP296463.

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
