# Peer review of "XA21-mediated resistance to Xanthomonas oryzae pv. oryzae is dose dependent"

_PeerJ, doi:10.7717/peerj.17323_

## Round 0.1 · original submission · Major Revisions

Reviewers 2 and 3 raised important issues that need to be addressed in this first round.

·

Basic reporting

This is a very impressive effort to understand the XA21-mediated defense in rice, especially regarding the impact on grain yield.

Experimental design

I think this paper follows the good procedures and use the methodology accordingly to achieve their objectives. The methods are well described and reproducible.

I was wondering just regarding the genotype choice (Kitaake - is a japonica line). Once rice indica and japonica present contrasting behavior against the pathogen, could be interesting to test indica also to do a comparative study. I am not sure if you have model genotypes available for that.

Validity of the findings

The results are valid and novel, contributing to the scientific community.

Additional comments

This is a very impressive effort to understand the XA21-mediated defense in rice, especially regarding the impact on grain yield. I was wondering just regarding the genotype choice (Kitaake - is a japonica line). Once rice indica and japonica present contrasting behavior against the pathogen, could be interesting to test indica also to do a comparative study. I am not sure if you have model genotypes available for that.

Reviewer 2 ·

Basic reporting

The manuscript uses an HA tag fused into a known rice-resistance gene XA21 to quantify a potential dosage effect on the resistance phenotype. XA21 has been described as a significant resistance gene towards rice bacterial blight. The encoded protein is a plasma membrane leucine-rich repeat receptor kinase of the NLR family. The authors provide evidence of increased resistance in a dosage-dependent manner and argue this demonstration depended on the HA-XA21 for protein quantification.
The work is self-contained and well-reported. Nonetheless, there are some missing pieces of information, such as the precise XA21 genomic region, since the genome has been sequenced—and the fragment size cloned in the vector for transformation. The authors mention a two-round of PCR amplification from the genome but no further details. The precise information on the HA tag position in the XA21 gene is missing. Finally, the agricultural traits analyzed are not sufficiently described in the material and methods nor in the result section to support the claim that agricultural traits were measured. Table S2 lists the primers, which are not cited in the main document. Which primer set is used in which amplification? This information can be added to the table in an extra column.

The results are well described except for the agronomical traits written in a rush and no quantitative information, only qualitative representation in Figure 3.

Experimental design

The experimental design is adequate for the hypothesis presented. Transgenic plant numbers and analyses are adequate. Gene expression profiling and bacteria inoculation are well-developed and meaningful.

Validity of the findings

The novelty of the findings in this manuscript is primarily based on two observations: first, a dosage-dependent expression of a resistance gene influences the scoring of tolerance to a bacterial pathogen, and second, no effect can be observed regarding the agronomical traits. Interpretation of the dosage dependence is solely based on the intensity of HA-XA21 accumulation. It does not evaluate the positional effect of T-DNA insertion in the genome of Kitaake or the potential genomic instability of the regeneration process using 2,4D. Quantitative analyses should accompany the agronomical traits study and the experimental procedure. No information regarding the traits chosen is given or supported by the literature.

The results presented do not support sentences in the discussion, which are in lines 268-272 and 280-281. There are not enough statistical supporting evidence for either.

Reviewer 3 ·

Basic reporting

In this manuscript, Zhang et al. describe the dosage effects of XA21 on its resistance to Xoo and rice yield using a significant number of transgenic lines. The immune receptor has been widely used for the study of genetic engineering and controversial findings have been reported. Therefore, careful characterization of multiple transgenic events at both DNA and protein levels in combination of important agronomic traits is important. The manuscript is well-written, and the data presented is convincing.

Experimental design

Overall, the experiments are well designed.

Suggestion:
The authors used “whole genome” sequencing to characterize the transgenic lines generated. It is not clear whether their sequencing data has covered the whole rice genome. Transgenes might be potentially missed due to incomplete genome sequencing of some transgenic lines. Thus, I propose that, as an alternative method, they may consider confirming the copy numbers of XA21 in the transgenic lines using Southern blot analysis.

Validity of the findings

The findings are validated.

Additional comments

1. As mentioned by the authors, Hao et al. reported a severe impact of XA21 on quality and yield in the japonica rice cultivar TP309. This lab has also previously generated XA21 lines in the TP309 background, discussion on the distinct findings should be included.
2. The quality of Fig.3A is poor. A better picture should be provided.

Reviewer 4 ·

Basic reporting

-I am satisfied the given English style and no need for language editing.
-Please check the references, following the journal style.

Experimental design

Overall, experimental design and methodology is quite well as per the experimental questions.
I have some questions regarding the whole genome sequecning need to be addressed in the article
1- how much the sequencing reads coverage, eg., 20x or 30x, etc.
2- what is the summary of data analysis and this need to be mentioned in the supplementary tables in order to other researcher can view this clearly before fetching the SRA data.

Validity of the findings

I am quite satisfied with the finding and this article really informative for the rice researchers.

---

## Round 0.2 · accepted · Accept

Authors addressed all issues raised by reviewer 2.

Reviewer 2 ·

Basic reporting

I want to let you know that the authors have provided an adequate revised version of the manuscript. The novel layout for Figure 3 and related text offers a better view and insight into the agronomic results. A few comments are provided to improve reading and clarity. As a minor suggestion, the paragraph from lines 73 to 85 refers to a single paper (Zhai et al. 2004), it needs not be recited as is. In this line, I would also suggest including in this paragraph the observations regarding indica and japonica that were provided in the response letter.

Experimental design

The difference between the transgenic line expressing Xa21 under the Ubi1 promoter and the one under its promoter remains intriguing. The experimental design adds a specific aspect to the extensive literature on the rice Xa21 resistance gene: the dosage effect.

Validity of the findings

As already reported in my previous review, the findings are relevant to the specific community.

Reviewer 3 ·

Basic reporting

The manuscript is well-written.

Experimental design

The experiments in this manuscript are well designed.

Validity of the findings

The findings are validated.

Additional comments

None